# Sensory Characteristics and Volatile Compounds of Herbal Teas and Mixtures of Bush Tea with Other Selected Herbal Teas of South Africa

**DOI:** 10.3390/foods9040496

**Published:** 2020-04-14

**Authors:** Florence Malongane, Lyndy Joy McGaw, Legesse Kassa Debusho, Fhatuwani Nixwell Mudau

**Affiliations:** 1Department of Life and Consumer Sciences, College of Agriculture and Environmental Sciences, University of South Africa, Private Bag X6, Florida 1710, South Africa; malonf@unisa.ac.za; 2Phytomedicine Programme, Department of Paraclinical Sciences, University of Pretoria, Private Bag X04, Onderstepoort, Pretoria 0110, South Africa; lyndy.mcgaw@up.ac.za; 3Department of Statistics, College of Science, Engineering & Technology, University of South Africa, Private Bag X6, Florida 1710, South Africa; debuslk@unisa.ac.za; 4Department of Agriculture and Animal Health, College of Agriculture and Environmental Sciences, University of South Africa, Private Bag X6, Florida 1710, South Africa; 5School of Agriculture, Earth and Environmental Sciences, University of Kwazulu Natal, P. Bag X01, Scottsville, Pietermaritzburg 3209, South Africa

**Keywords:** herbal tea, GC-MS, flavour, volatile compounds, descriptive sensory evaluation

## Abstract

South Africa has a traditional heritage of using indigenous herbal teas, and the demand for herbal teas motivated by the functional health benefits has far exceeded global supply. This has led to worldwide interest in the sensory characteristics and volatile compound characterisation of herbal drink formulations. The objective of this study was to investigate the descriptive sensory analysis and volatile compounds of bush, special, honeybush and rooibos tea and the blend of bush tea with special, honeybush and rooibos, respectively. The trained sensory panel scored each tea sample for aroma, taste, aftertaste and mouthfeel attributes using sensory evaluation practices. Compound identification was performed by gas chromatography connected to a mass spectrometer (GC-MS). The results of the study demonstrated that rooibos and honeybush tea had an overall sweet-caramel, honey-sweet, perfume floral and woody aroma while bush tea and special tea depicted green-cut grass, dry green herbal and astringent/dry mouth feel. The GC-MS analyses depicted the following compounds 2-furanmethanol, 2-methoxy-4-vinylphenol, D-limonene, dihydroactinidolide, linalool, (E,E)-2,4-heptadienal, and phytol. The blending of bush tea with rooibos and honeybush tea toned down its astringent mouth feel. Compounds identified in this study may be useful markers for potential herbal tea sensory characteristics.

## 1. Introduction 

South Africa has a traditional heritage of using indigenous herbs to make tea, with the most popular being rooibos tea (*Aspalathus linearis)*, honeybush tea (*Cyclopia* species), bush tea (*Athrixia phylicoides*) and special tea (*Monsonia burkeana)* [1]. Rooibos tea has been commercialised and is currently distributed in local and international markets such as Germany (31%), Netherlands (16%), Japan (15%), the United Kingdom (11%), United States of America (7%) and the remaining 20% is exported to India and Sri Lanka [2]. Similarly, honeybush tea comprising a blend of *Cyclopia* species, usually *C. intermedia*, *C. genistoides* and *C. subternata*, is also exported to the international market [3]. However, bush and special tea, the herbal teas used in former homeland areas of South Africa, have not yet been fully explored, mainly due to the underproduction and unavailability of these teas in the formal market, although they are traded in the informal market in large quantities [4]. Both these teas possess good antibacterial and anti-oxidant properties [5,6]. Increased knowledge of these teas will increase diversity and create more opportunities for a sustainable herbal tea market in South Africa. 

Research on descriptive sensory and volatile compounds analysis has focused on rooibos and honeybush tea due to their worldwide consumer demand and with the aim of producing appealing competitive products [2,3]. There are currently no reported results on the sensory analysis of bush and special tea, nor has any study been carried out on the sensory analysis of bush tea blended with special, rooibos and honeybush teas. The sensory attribute characteristics of fermented rooibos tea are honey, woody and herbal-floral flavours, a slightly sweet taste, astringent, caramel flavour and a sweet-associated fruity flavour [7]. Furthermore, Jolley et al. [2] reported aroma characteristics of rooibos as fynbos-floral, honey, hay/dried grass and rooibos-woody. The sensory characteristics of fermented honeybush tea are listed as fynbos-sweet, fynbos-floral, woody, sweet and slightly astringent mouthfeel [3]. The analysis of the selected tea samples will enable the profiling of the sensory characteristics of both the commercial and non-commercial teas in order to deduce the consumer acceptability of the non-commercial teas. Additionally, the blending of bush tea with other herbal teas could increase the sensory appeal and strengthen its potential for commercialisation. 

The consumption of herbal teas has increased in South Africa with the growth in production of rooibos and honeybush teas, as well as other non-commercial herbal tea species such as bush tea [8]. Herbal teas are infusions made from fruits, leaves, flowers, roots and stems of plants and are intended for oral aqueous consumption [9]. The increased use of herbal teas is attributed to their sensory properties (aroma and taste) and perceived health benefits [8].

The combination of sensory and chemical analysis data can provide more insight into product characteristics and useful markers for tea sensory characteristics [10]. However, data describing the chemical compositions and sensory attributes of both bush and special tea are lacking. Therefore, the objective of this study was to determine sensory and volatile compounds following the blending of these selected teas with known commercial herbal teas.

## 2. Material and Methods 

### 2.1. Plant Materials 

Seven tea samples including bush tea, special tea, honeybush tea and rooibos tea as well as three blends of bush tea in a 1:1 ratio (bush + rooibos tea, bush + honeybush tea and bush + special tea) were characterised for their sensory attributes and volatile compound constitution. Bush tea samples were collected from the Agricultural Research Council (Pretoria, South Africa), special tea leaves from Hartbeespoort (North West Province, South Africa) while rooibos (*Aspalathus linearis*) and honeybush (*Cyclopia* species) tea samples were bought from a local market in Florida, Johannesburg (Gauteng Province, South Africa). The four South African herbal teas were selected on the basis that they were locally produced.

### 2.2. Descriptive Sensory Evaluation 

Descriptive sensory analysis (DSA) was conducted in the Sensory Analysis Unit of the Agricultural Research Council (ARC) in Irene, South Africa. The method includes determining product descriptors, examining the sensory perceived attributes and measuring the intensities of those attributes by trained panellists whose roles were more like analytical tools or instruments in that they described what attributes were perceived and how strong/intense they were. 

#### 2.2.1. Procedure-Preparation

The leaves of special tea and bush tea were oven-dried at 50 °C for 48 h and then ground to a coarse powder resembling un-ground rooibos tea. The tea powders were stored in glass jars and in a dark place until further analysis. Bush tea was blended with the other three teas at a ratio of 1:1 (*w*/*w*). Twenty grams (20 g) of each tea sample was placed in porcelain kettles pre-heated to 70 °C in an oven to avoid rapid cooling of water after which 1 L of boiling distilled water was poured into the porcelain tea pot. Tea was allowed to brew for 5 min according to the descriptions of Welna et al. [11] while being kept warm in a heated oven (70 °C). Brewed tea was sieved and poured into warmed porcelain cups and covered immediately with Petri dish lids. Each panel member was served immediately with 100 mL of brewed tea per session. Tasting of the tea samples was randomised over four days in sessions lasting approximately 10 min so that each product had an equal opportunity to be tasted first. Panellists evaluated products in separate tasting booths to reduce distraction and panellist interaction and to ensure uninterrupted and unbiased individual responses. All products were coded with a three-digit random code and presented blindly to the panellists. Tea was evaluated under red light conditions to mask the colour differences. Water biscuits from a local market and distilled water at room temperature were served as palate cleansers in between evaluation sessions as previously described [12,13]. Care was taken to ensure that each sample was identically treated with respect to the volume served and serving temperature of each replication of the tea samples. 

The actual tasting was conducted according to the method described by Owuor [14]. Briefly, the testing was carried out by sucking, rather than sipping, so that the liquor is drawn to the back of the mouth, on an inward breath, and up to the olfactory nerve in the nose. The liquor was then swished backwards and forwards and brought into contact with the tongue, palate, and other areas of the mouth where sensory receptors are located. In this way, the panellists were able to feel, taste, and smell the liquor virtually simultaneously and were thus able to determine its briskness, strength, body and flavour. The liquor was taken directly from the cup and discarded into a disposable cup after tasting. 

#### 2.2.2. Taste Panel Procedures

Ten (10) experienced trained panellists from the Agricultural Research Council – Animal Production Institute (Irene, Pretoria, South Africa) who had previously participated in the descriptive sensory evaluation of beverages, meat and vegetables, were selected for the project. The ten trained panellists were a representative sample to produce good quality data [15]. The panellists were informed not to use perfumed cosmetics, to avoid exposure to foods and/or fragrances at least an hour before the evaluation sessions and to avoid burning their oral cavity with any liquid as described by Lee et al. [16]. Prior to DSA, the panellists discussed sensory attributes of the seven tea samples during the preliminary four sessions, each lasting 120 min until they agreed on 26 sensory descriptors (Table 1). A category rating scale, with zero (0) denoting no perceived flavour (e.g., no sweet-caramel, maple syrup aroma) and seven (7) denoting the most intense condition (e.g., extremely intense sweet-caramel, maple syrup) was constructed and used to evaluate the different samples. 

### 2.3. Gas Chromatography-Mass Spectrometry

#### 2.3.1. Tea Samples 

The water extraction method was used for the preparation of all tea samples. In brief, 4 g of each tea sample was placed in a 50 mL centrifuge tube followed by the addition of 40 mL of boiled (100 °C) deionised water to each sample as previously described [17]. The mixtures were vigorously shaken, sonicated and then centrifuged; each step lasted 10 min. The resultant liquid was filtered into a beaker through Whatman No. 1 filter paper (11 µm pore size). The extraction was repeated using 20 mL and then 10 mL of boiled deionised water on the same plant material. The filtered tea extracts were stored at −80 °C overnight, and then freeze-dried. All tea extracts were reconstituted in methanol (Sigma Aldrich) and dichloromethane (Sigma Aldrich) at a ratio of 1:1 to a concentration of 100 mg/mL. Tea supernatants were placed directly into 2 mL amber screw-top vials. The samples for the gas chromatography-mass spectrometer (GC-MS) were prepared and analysed in triplicate. 

#### 2.3.2. Gas Chromatography-Mass Spectrometry 

The samples were analysed by gas chromatography (Agilent Technologies 7890B) that was equipped with a LECO, Pegasus 4-D GC X GC-TOF MS detector and Gerstel multi-purpose sampler (GMbH and Co. KG, 45473 Mülheim an der Ruhr, Germany). The oven was equipped with an Agilent HP-5MS column (30 m × 0.25 mm × 0.25 µm film thickness) fused silica capillary column. 

The Helium carrier gas (percentage purity > 99.999%) was maintained at a constant flow of 1.5 mL/min. A 1.5 µL splitless injection volume was used with an inlet temperature of 250 °C. The chromatographic separation was achieved by using the following temperature program. The temperature was initially set at 35 °C (held for 1 min), and then increased to 142 °C at a ramp rate of 5 °C/min (held for 3 min), then increased further to 240 °C at 5 °C/min. The transfer line temperature was maintained at 225 °C. The MS ion source temperature was 200 °C with filament bias of −70 voltage. The MS mass range units were 40 to 550 atomic mass units (amu), with the acquisition rate of 10 spectra per second while the detector voltage was 1670. The total GC run time was 90 min. 

The mass spectrometer was operated in full-scan mode, and the peak area was determined by ChemStation software (Agilent Technologies). Tentative identifications of volatile compounds were based on matching mass spectra of unknowns with those of the reference database (NIST Mass Spectral Data 98 edition) according to the descriptions of Wu et al. [18].

### 2.4. Statistical Techniques

The data obtained from the sensory panel were analysed using the repeated measures analysis of variance (ANOVA) to compare all the sensory attributes measured among tea samples. Samples for GC-MS were prepared and analysed in triplicate. The comparison of each GC-MS measurement (variables) among tea samples was done utilizing one-way ANOVA. The Tukey’s (Honest significant difference) HSD test was applied for pairwise means comparison of tea samples, at a 5% level of significance. Principal component analysis (PCA) was used in this study to determine whether there were differences in levels of descriptive sensory attributes among the seven tea samples. A partial least square regression analysis was performed using the covariance matrix to determine the relationship between the data from the sensory panel and the GC-MS data. All the analyses were done using JMP^®^, Version 13 (SAS Institute Inc., 2016).

### 2.5. Ethics Approval 

The study was reviewed in compliance with the Unisa Policy on Research Ethics by the College of Agriculture and Environmental Sciences (CAES) research ethics review committee. The approval was granted before the study began. 

## 3. Results and Discussion

### 3.1. Descriptive Sensory Analysis 

The aroma attributes such as sweet-caramel, honey, green-cut grass, cooked spinach, dry green herbal, earthy, perfume-floral and woody-cinnamon; the taste such as, bitter-quinine, sweet-caramel, honey, rooibos, green-cut grass, cooked spinach, dry green herbal, earthy, fruity-peach, perfume-floral, woody-cinnamon and medicinal; the aftertaste including, bitter, green-cut grass, cooked spinach, dry green herbal, woody-cinnamon and earthy; and the mouthfeel such as astringent/dry, were evaluated. Some aroma attributes were similar to those found in rooibos, honeybush and black tea, which were woody, honey and green. The mean scores and the significant difference for all 26 attributes are described in Table 2. Appendix A contains raw data from descriptive sensory evaluation.

#### 3.1.1. Aroma 

According to the results given in Table 2, sweet-caramel, honey-sweet, perfume-floral and woody aromas were more prominent in the rooibos and honeybush teas, though at a low intensity. Theron et al. [3] reported a fynbos-floral, fynbos-sweet, cinnamon and woody aroma in honeybush tea, which is in agreement with the findings of the present study. The aroma characteristics of rooibos previously discussed were rooibos woody, fynbos-floral and honey [2]. Green-cut grass, dry green herbal were very low in rooibos and honeybush teas, but noticeably higher in bush tea, special tea and all three bush tea blends. Special tea had the most intense cooked spinach aroma and as a result scored high in bush-special tea blend. There was only a hint of earthy boiled potatoes and damp potting soil aroma in all seven samples. There are no previously reported aroma characteristics of special tea and bush tea, nor of the blend of bush tea, making this the first study to report such results. 

#### 3.1.2. Taste Profile 

The taste profiles of all teas are presented in Table 2. The strongest bitter-quinine, caffeine flavour was detected in bush, special tea and the bush tea and special tea blend. High intensities for bush tea also increased scores for the bush and honeybush tea blends and bush and rooibos tea blends. In the current study, bush and special tea were not fully fermented compared to rooibos and honeybush teas, which underwent a quality control fermentation process. A previous study demonstrated that the fermentation process reduces the bitterness in tea [19], and such could be the case with the bitter taste found in bush and special tea. In the production of rooibos tea, the process includes the formation of sweet, apple-like or honey caramel, after which the fermentation process is halted, thus ensuring the characteristics of sweet in the final product [20]. The sweet-caramel and typical honey-sweet flavours were similar to the aroma more prominent in rooibos and honeybush tea, and consequently increased the scores of their respective blends with bush tea. In contrast, there was only a hint of honey-sweet typical honey flavor in the bush tea, special tea and the bush and special tea blend. Rooibos flavour was the highest in the rooibos and honeybush teas, followed by bush-rooibos and bush-honeybush tea blends, and was the least intense in bush tea and special tea. Blending bush tea with rooibos and honeybush teas increased the intensity of sweet-caramel, honey-sweet and rooibos, providing a modifying factor to the bitter taste of bush tea alone. Bush tea was the only tea with a high score of intense green-cut grass taste. Special tea and the special-bush tea blend were the only two teas with an intense cooked spinach and medicinal flavour. According to the results, the strongest dry green herbal taste was detected in bush tea, special tea, bush-rooibos tea blend, bush-honeybush tea blend and bush-special tea blend. Overall, for all seven teas, the earthy boiled potatoes and woody cinnamon were of a low intensity. The perfume—floral lavender—was only intense in the honeybush tea and the bush-honeybush tea blend. This is the distinguishing factor of honeybush compared to other teas in the study. 

#### 3.1.3. Aftertaste and Mouthfeel of Seven Teas

The bitter and green-cut grass aftertaste was more prominent in bush tea, special tea and all bush tea blends, similar to the results in the tasty flavour. The cooked spinach aftertaste was more intense in the special tea and bush-special tea blends. Again, as with the taste results, the strongest dry green herbal aftertaste was detected in bush tea, special tea and in all bush tea blends. Rooibos and honey bush tea were the only two teas with a more woody-cinnamon aftertaste, which was in agreement with the findings of a previous study on honeybush tea [2]. Overall, for all seven teas, the earthy boiled potatoes aftertaste was very low (1.18–1.92) as was also the case with the taste attributes. The astringent/dry mouth feel was very weak in the rooibos and honeybush teas, in contrast with bush tea, special tea and bush-special tea blends that had the most intense astringent/dry mouth feel of the seven teas. The strong astringent aftertaste could be a negative attribute and might contribute to consumer aversion towards bush tea and special tea [19]. Blending of bush tea with rooibos tea or honeybush tea could act as a de-bittering process to reduce the bitterness of bush tea. 

#### 3.1.4. Colour

The results regarding the colour scores of tea samples are depicted in Table 2 which revealed a significant variation (*p* < 0.0001) among different tea samples. In a similair manner, the orange colour was more described in honeybush and rooibos tea and the blend of bush tea and rooibos. Bush tea showed an intense green colour followed by special tea and the blend of bush tea and special tea. Blending bush tea with rooibos and honeybush tea reduced the green colour depicted in bush tea, positioning bush tea as a likeable tea.

Figure 1 shows the results of the principal component analysis (PCA) of all tea samples and bush tea blend attributes on axis 1 (76.08%) and axis 2 (13.30%). The PCA (F1) biplot grouped honeybush and rooibos tea in the same cluster while the blend of bush tea with honeybush or rooibos was grouped together showing that both rooibos and honeybush share similar characteristics. Similarly, special tea and the blend of special tea with bush tea shared the same clustering as indicated by PCA 1 while bush tea was clustered alone. In contrast, PCA 2 (13.30%) showed a distinct clustering of rooibos, honeybush, the rooibos and bush tea blend and the bush and honeybush tea blend, the other grouping being of bush, special tea and the bush and special tea blend.

PCA 1 shows that attributes such as ATwoody, Tsweet, Thoney, Ahoney, Twoody, Tperfume, were more profound in rooibos and honeybush tea. Blending with bush tea resulted in a mild flavour of Aperfume, Awoody, Asweet and Trooibos. Bush tea was mostly characterised by colour green, Aherbal, ATherbal, Therbal, ATgreen, Agreen and Tgreen while special tea had a brown colour, Tspinach, Aspinach, ATspinach, ATbitter, Tbitter, ATearthy, Aearthy and MFdry. 

### 3.2. GC-MS Identification of Volatile Compounds in Selected Teas 

Seven tea samples were analysed and the representative total ion chromatogram (TIC) for each of these teas is presented in Figure 2 and Table 3. Raw data sourced from gas chromatography-mass spectrometer (GC-MS) is found under Appendix A. A total of 58 volatile compounds in the tea extracts were detected by GC-MS. The identity of the compounds was determined by comparing their spectra to those of known compounds in the Mass Spectral Library (NIST08, NIST08s, FFNSC1.3). Compounds were selected for consideration on the basis of having a similarity cut-off of 800 and above. The identified volatile compounds and their relative area percentage, mean values and standard error are summarised in Table 3. Volatile compounds from different chemical groups such as terpenoids, alcohols, hydrocarbons, aldehydes, acids, furans, phenols, esters and ketones are known to influence the typical flavour of teas [21]. 

The volatile compounds (-)-carvone, (-)-α-copaene, 2-furanmethanol, 2-methoxy-4-vinylphenol, acetic acid, α-myrcene, α-ocimene, apocynin, caryophyllene, d-limonene, geraniol, linalool, methyl salicylate, phenylethyl alcohol, phytol, piperonal, squalene, vanillin, hydroxyacetone, beta-ocimene, 2-methyl butyraldehyde, actinidiolide, dihydroactinidolide, terpinolene, (E,E)-2,4-heptadienal, maltol, cis-2,6-dimethyl-2,6-octadiene, p-allylphenol, trans-4-propenylsyringol, diacetone alcohol, cadina-1(10),4-diene, trans-verbenol, trans-geranylgeraniol, humulene, maltol, cis-ocimenol, and acetoin were tentatively identified in the tea samples. 

(-)-Carvone (GC1), (-)-α-copaene (GC3), caryophyllene (GC11), piperonal (GC29) trans-verbenol (GC52), trans-geranylgeraniol (GC53), and humulene (GC54) were among the compounds detected in special tea for which no study has previously been done regarding the GC-MS analysis of its volatile compound composition. 2-Furanmethanol (GC5) (a flavouring ingredient) was found in all teas except in the bush-special tea blend. This compound has previously been reported in oolong, green and black tea [22]. 2-Methoxy-4-vinylphenol (GC6) and dihydroactinidolide (GC35) were detected in all tea samples, albeit with slight differences in their relative abundance in the teas. 2-Methoxy-4-vinylphenol was identified as a key contributor to aroma compounds in tea beverages [23] while dihydroactinidolide was detected in Assam black tea and Darjeeling black tea [24]. Honeybush tea was shown to have α-myrcene (GC8), which was subsequently detected also in the blend of bush and honeybush tea. This compound was previously identified in alcoholic beverages [25]. However, β-myrcene, the trans arrangement of myrcene, was previously identified in honeybush tea [26]. Geraniol (GC14), and cis-ocimenol (GC51), were detected only in honeybush tea. 

Linalool (GC15) and D-limonene (GC12) were tentatively identified in bush and honeybush teas and the blend of both teas, congruent with the findings of Ntlhokwe et al. [27]. Bush tea was the only tea which had 2-methyl butyraldehyde (GC30), cis-2,6-dimethyl-2,6-octadiene (GC31), p-allylphenol (GC32), trans-isoeugenol (GC44), trans-4-propenylsyringol (GC45), diacetone alcohol (GC48), and Cadina-1(10),4-diene (GC42). Beta-Ocimene, (GC29) and Gamma-Terpinene (GC46) were previously detected in bush tea [28]. The distinct compounds found in rooibos tea were acetoin (GC57) and maltol (GC58). 

Phenylethyl alcohol was detected in honeybush and rooibos teas and also in the blends of bush-honeybush and bush-rooibos teas. These results confirmed the findings of Marnewick (2009) [29] who identified phenylethyl alcohol in both teas. Phytol (GC18) was detected in all tea samples though there were slight variations in its relative abundance in the teas. Previous studies reported its presence in green, oolong and Pu-erh tea [21]. Squalene (GC21), pseudoionone (GC547) and glycerin (GC34) were the only compounds detected in bush tea and special tea.

**Table 3 foods-09-00496-t003:** Compounds detected and concentrations using a gas chromatography-mass spectrometer (GC-MS) in seven tea samples.

Code	Name of Compounds	RT (m)	Odour Description	Bush Tea	Honeybush Tea	Special Tea	Rooibos Tea	Bush: Honeybush	Bush: Special	Bush: Rooibos
GC1	(-)-Carvone	18.39	Spearmint-like herbal odour [30]	ND	ND	0.025 ± 0.007	ND	ND	ND	ND
GC2	(E)-1-(2,3,6-trimethylphenyl)buta-1,3-diene (TPB, 1)	22.77	-	0.039 *	ND	0.050 *	0.100 *	0.0022 *	0.122 *	ND
GC3	α-Copaene	21.92	Woody, spicy [31]	ND	ND	0.11 ± 0.04	ND	ND	ND	ND
GC4	1-Pentanol	4.25	-	ND	ND	0.187	ND	ND	0.157	ND
GC5	2-Furanmethanol	6.86	Weak, creamy, burnt sugar [32]	0.054 ± 0.0.022 ^a^	0.266 ± 0.083 ^ab^	0.111 ± 0.013 ^bc^	0.0126 ^bc^*	0.394 ± 0.131 ^a^	ND	0.116 ± 0.029 ^bc^
GC6	2-Methoxy-4-vinylphenol	20.31	Smoky [33]	0.308 ± 0.066 ^c^	0.422 ± 0.149 ^c^	1.677 ± 2.692 ^bc^	6.834 ± 0.989 ^a^	0.460 ± 0.052 ^c^	0.437 ± 0.049 ^c^	3.419 ± 0.124 ^b^
GC7	Acetic acid	2.51	Acidic [34]	0.421 ± 0.263 ^c^	0.919 ± 0.439 ^bc^	0.405 ± 0.260 ^c^	2.974 ± 0.486 ^a^	2.728 ± 0.811 ^a^	0.638 ± 0.027 ^c^	1.493 ± 0.287 ^b^
GC8	α-Myrcene	10.85	Sweet [35]	0.121 ± 0.022 ^c^	2.569 ± 0.179 ^a^	ND	ND	1.411 ± 0.304 ^b^	0.225 ± 0.0.072^c^	0.115 ^c^*
GC9	α-Ocimene	12.66		0.047 ^c^*	1.316 ± 0.172 ^a^	ND	ND	0.774 ± 0.049 ^b^	0.168 ^c^*	ND
GC10	Apocynin	24.79	Sweet, somewhat vanilla-like [36]	0.042 *^c^	0.249 ± 0.011 ^b^	ND	0.546 ± 0.113 ^a^	0.081 ^bc^*	ND	0.262 ± 0.044 ^b^
GC11	Caryophyllene	23.04	-	ND	ND	0.780 ± 0.280 ^a^	ND	ND	0.044 ^a^*	ND
GC12	D-Limonene	11.94	Citrus, lemon. [37]	0.158 ± 0.056 ^d^	1.969 ± 0.066 ^a^	0.058 ± 0.011 ^d^	ND	1.241 ± 0.134 ^b^	0.303 ± 0.043 ^c^	0.099 ± 0.016 ^d^
GC13	Furfural	6.26	-	0.042 ± 0.005 ^cd^	0.882 ± 0.068 ^a^	0.040 ± 0.009 ^cd^	ND	0.216 ± 0.048 ^b^	0.041 ± 0.044 ^cd^	0.030 ± 0.008 ^d^
GC14	Geraniol	18.81	Floral, woody [38]	ND	0.183 ± 0.020 ^a^	ND	ND	0.113 ± 0.009 ^b^	ND	ND
GC15	Linalool	14.28	Fruity, floral [39]	0.015 ± 0.003 ^c^	0.161 ± 0.015 ^a^	ND	ND	0.106 ± 0.020 ^b^	ND	ND
GC16	Methyl salicylate	16.96	Sweet, characteristic wintergreen [36]	ND	0.038 ± 0.008 ^a^	ND	0.101 ± 0.005	0.153 ± 0.113 ^a^	ND	ND
GC17	Phenylethyl alcohol	14.66	Floral [38]	ND	0.502 ± 0.081 ^b^	ND	0.919 ± 0.288 ^a^	0.206 ± 0.020 ^bc^	ND	0.118 ± 0.081 ^c^
GC18	Phytol	57.37	Sweet, Floral [38]	0.745 ± 0.136 ^b^	1.286 ± 0.136 ^b^	6.848 ± 2.991 ^a^	1.377 ± 0.203 ^b^	2.602 ± 0.304 ^b^	5.174 ± 0.651 ^a^	1.407 ± 0.070 ^b^
GC19	Piperonal	20.78	Cherry, vanilla, sweet anisic [36]	ND	ND	0.058 ± 0.041 ^a^	ND	ND	0.274 ± 0.267 ^a^	ND
GC20	Salicylic acid	20.53	-	0.067 ^b^*	0.152 ^b^*	0.123 ± 0.040 ^b^	1.449 ± 0.446 ^a^	0.092 ± 0.032 ^b^	ND	0.380 ^b^*
GC21	Squalene	108.63		0.638±.0229 ^ab^	ND	1.279 ± 0.719 ^a^	ND	0.084 ± 0.036 ^b^	0.611 ± 0.066 ^ab^	0.592 ± 0.395 ^ab^
GC22	Vanillin	22.58	Vanilla-like, sweet [23]	0.031 ^b^*	ND	ND	0.710 ± 0.184 ^a^	0.091 ^b^*	0.091 ± 0.042 ^b^	0.389 ± 0.043 ^ab^
GC23	Hydroxyacetone	3.77	-	0.176 ± 0.185 ^ab^	0.131 ^b^*	ND	0.487 ± 0.128 ^ab^	0.237 ± 0.082 ^ab^	0.677 ^a^*	0.219 ^ab^*
GC24	Propiophenone	3.94	-	0.010 ± 0.002 ^b^	0.068 ± 0.038 ^a^	0.045 ± 0.001 ^ab^	0.016 ^b^*	0.022 ± 0.016 ^b^	0.025 ± 0.020 ^b^	0.031 ± 0.026 ^ab^
GC25	Butyrolactone	8.45	-	0.263 ± 0.160 ^a^	0.134 ± 0.003 ^a^	ND	0.098 ± 0.010 ^a^	0.260 ± 0.093 ^a^	ND	ND
GC26	5-Methyl-2-furancarboxaldehyde	3.94	-	0.027 ± 0.021 ^b^	0.705 ± 0.436 ^a^	0.084 ± 0.002 ^b^	0.091 ± 0.002 ^b^	0.457 ± 0.297 ^ab^	0.085 ± 0.041 ^b^	0.033 ± 0.004 ^b^
GC27	2-Methylpyrazine	6.02	-	0.030 ± 0.012 ^a^	ND	0.037 ± 0.014 ^a^	ND	0.023 ^a^*	0.024 ± 0.014 ^a^	0.020 ± 0.020 ^a^
GC28	4′-Methylacetophenone	16.65	-	0.043 ± 0.007 ^c^	0.030 ± 0.000 ^c^	0.239 ± 0.085 ^a^	0.072 ± 0.019 ^bc^	0.029 ± 0.004 ^c^	0.131 ± 0.009 ^b^	0.048 ± 0.030 ^c^
GC29	beta-Ocimene	12.63	[28]	0.101 ^a^*	ND	ND	ND	0.051 ± 0.060 ^a^	0.074 ± 0.037 ^a^	0.040 ± 0.026 ^a^
GC30	2-methylbutyraldehyde	3.91	Fruity [39]	0.205 ± 0.015 ^a^	ND	ND	ND	0.158 ± 0.137 ^a^	0.424 ^a^*	ND
GC31	cis-2,6-Dimethyl-2,6-octadiene	11.18	-	0.129 ± 0.022 ^ab^	ND	ND	ND	0.061 ^b^*	0.217 ^a^*	ND
GC32	p-Allylphenol	21.11	-	0.039 ± 0.021	ND	ND	ND	ND	ND	ND
GC33	1,1,6-Trimethyl-1,2-dihydronaphthalen	21.28	-	0.075 ± 0.032 ^a^	0.082 ± 0.108 ^a^	0.089 ± 0.02 ^5a^	0.0602 ± 0.030 ^a^	0.091 ± 0.052 ^a^	0.112 ± 0.019 ^a^	0.129 ± 0.111 ^a^
GC34	Glycerin	13.98	-	0.752 ± 0.374 ^b^	ND	0.339 ± 0.340 ^b^	ND	3.464 ^a^*	0.464 ± 0.582 ^b^	0.120 ± 0.047 ^b^
GC35	Dihydroactinidolide	25.82	Floral rose like [24]	0.172 ± 0.085 ^d^	0.321 ± 0.043 ^d^	2.023 ± 0.199 ^a^	0.938 ± 0.173 ^b^	0.382 ± 0.037 ^d^	1.174 ± 0.233 ^b^	0.648 ± 0.089 ^c^
GC36	1,1,6-trimethyltetralin	21.38		0.017 ± 0.012 ^b^	0.080 ± 0.066 ^ab^	0.012 ± 0.008 ^b^	0.119 ± 0.054 ^a^	0.029 ± 0.010 ^b^	0.055 ± 0.057 ^ab^	0.018 ± 0.012 ^b^
GC37	2-Hydroxy-2-cyclopenten-1-one	8.85	-	0.030 ± 0.023 ^d^	0.986 ± 0.159 ^a^	0.188 ± 0.055 ^cd^	0.391 ± 0.118 ^b^	0.287 ± 0.189 ^bc^	0.065 ± 0.062 ^d^	0.063 ± 0.052 ^d^
GC38	α-Angelica lactone	7.17	-	0.039 ± 0.016 ^bc^	0.986 ± 0.159 ^ab^	0.188 ± 0.055 ^a^	0.391 ± 0.118 ^bc^	0.287 ± 0.189 ^a^	0.087 ± 0.054 ^ab^	0.063 ± 0.052 ^ab^
GC39	Dehydro-beta-ionone	24.64	-	0.003 ^c^*	0.043 ± 0.001 ^bc^	0.050 ^bc^*	0.179 ± 0.035 ^a^	0.035 ± 0.007 ^c^	0.087 ^b^*	0.054 ± 0.033 ^bc^
GC40	Terpinolene	13.80	37	0.376 ± 0.406 ^bc^	0.906 ± 0.047 ^a^	ND	ND	0.538 ± 0.022 ^b^	0.151 ± 0.085 ^cd^	0.036 ± 0.005 ^d^
GC41	Cadina-1(10),4-diene	25.68	-	0.010 ± 0.007	ND	ND	ND	ND	ND	ND
GC42	3-Thujene	11.15	-	0.050 ^a^*	0.147 ± 0.119 ^a^	ND	ND	0.080 ± 0.003 ^a^	ND	ND
GC43	Methyl glycolate	2.70	-	0.205 *	ND	ND	ND	ND	ND	0.125 *
GC44	trans-Isoeugenol	23.78	-	0.090 ± 0.071	ND	ND	ND	ND	ND	ND
GC45	trans-4-PROPENYLSYRINGOL	31.90	-	0.020 ± 0.023	ND	ND	ND	0.222 *	ND	ND
GC46	Gamma-Terpinene	12.89		0.033 ± 0.002	0.209 ± 0.077	ND	ND	ND	ND	ND
GC47	Pseudoionone	39.33	-	0.791 ± 0.098^bc^	ND	1.317 ^ab^*	ND	0.643 ^c^*	1.615 ± 0.179 ^a^	0.941 ^bc^*
GC48	Diacetone alcohol	7.72	-	0.016 *	ND	ND	ND	ND	ND	ND
GC49	(E,E)-2,4-heptadienal	11.47	fatty, nutty, hay [24,40,41]	0.010 ^b^*	0.027 ± 0.023 ^b^	ND	0.162 ± 0.038 ^b^	0.723 ± 0.301 ^a^	0.239 ^b^*	0.219 ± 0.0217 ^b^
GC50	cis-Ocimenol	16.17	-	ND	0.346 ± 0.374	ND	ND	ND	ND	ND
GC51	trans-Verbenol	15.53	-	ND	ND	0.325 ± 0.118	ND	ND	ND	ND
GC52	6,10,14-Trimethylpentadecan-2-one	39.33	-	0.651 ^b^*	0.310 ^c^*	ND	0.719 ± 0.066 ^ab^	0.606 ± 0.094 ^b^	0.567 ± 0.005 ^b^	0.865 ± 0.082 ^a^
GC53	trans-Geranylgeraniol	48.81	-	ND	ND	0.292 ± 0.033	ND	ND	ND	ND
GC54	Humulene	23.88	-	ND	ND	0.296 ± 0.342	ND	ND	ND	ND
GC55	Butyrovanillone	27.66	-	0.007 ^a^*	0.414 ± 0.470 ^a^	0.009 ± 0.005 ^a^	0.690 ± 0.629 ^a^	0.071 ± 0.029 ^a^	0.021 ± 0.017 ^a^	0.480 ± 0.303 ^a^
GC56	Homovanillyl alcohol	26.03	-	ND	ND	ND	0.243 ^a^*	0.689 ± 0.466 ^a^	ND	ND
GC57	Acetoin	3.88	-	ND	ND	ND	0.252 ± 0.094	ND	ND	ND
GC58	Maltol	14.65	Caramel [42]	ND	ND	ND	0.319 ± 0.012^a^	0.029 ± 0.007 ^b^	ND	ND

Comparison of measured peak areas for annotated compounds in selected herbal teas with tentative identification. Level of significance was determined. a,b,c, Different superscript letters signify statistically significant differences in the relative amounts of the compounds in each tea. ND. Not detected. RT. Retention time. ^a,b,c,d^ are taken to signify statistically significant differences in the relative intensity of each tea. * Indicate that the experiment detected one value out of the three repitations.

### 3.3. Relationships between Descriptive Sensory and Volatile Compound Analyses in Selected Teas 

The Partial Least Squares Regression (PLSR) biplot of the selected herbal teas, descriptive sensory attributes and volatile compounds is shown in Figure 3. Special tea lies slightly outside the confidence ellipse, indicating that it might be influential in the PLSR analysis. This trend is followed by honeybush tea. When variables on the map were proximal, they represent a closer relationship among the sensory attributes and volatile compounds. The variable importance plot showed 25 predictors with Variable Importance in Projection (VIP) values exceeding 0.8, indicating that 25 of the 26 sensory attributes were influential in determining the two factors. Dependent variables, as explained by the model, are reflected in the fact that, except the points for (-)-carvone (GC1), (E)-1-(2,3,6-trimethylphenyl)buta-1,3-diene (GC2), butyrolactone (GC25), 2-methyl butyraldehyde (GC30), and diacetone alcohol (GC48), which are within the 25% circle, the others are near either the 50%, 75% or the 100% circle.

Volatile compounds are key contributors to tea flavour [33]. The typical flavour of each tea type is influenced by plant morphology and the degree of fermentation [41,43]. 2-furanmethanol (weak, creamy, burnt sugar), d-limonene (citrus, lemon), α-myrcene (sweet), geraniol (floral, woody), linalool (fruity, floral), phenylethyl alcohol (floral) and apocynin (sweet) were found in honeybush tea and were correlated with the taste profile perfume and woody, woody aftertaste and sweet aroma. Honeybush tea showed a high score on sweet taste and had a high relative abundance of α-myrcene, linalool and apocynin. Phenylethyl alcohol was reported in honeybush tea and was shown to impart a sweet taste [29]. The sensory evaluation did not include burnt sugar or lemon in the assessment; therefore, any volatile compounds associated with such attributes were not compared. 

(E,E)-2,4-heptadienal (nutty, hay) and maltol (caramel) were located in the quadrant between rooibos and honeybush. Though acetic acid was a predominant compound in all tea samples, its relative abundance was higher in rooibos, honeybush tea and the bush-honeybush tea blend. Rooibos tea was associated with phenolic compounds, 2-methoxy-4-vinylphenol (smoky) and vanillin (vanilla-like, sweet). The flavour component and the volatile compounds of any tea are greatly influenced by post-harvest activities such as fermentation, which contribute to the sweet taste in tea [41,43]. The sweet caramel taste in fermented rooibos tea and honeybush tea could be as a result of fermentation. 

Squalene had a negative correlation with all tea samples except for special tea. Special tea was coupled together with a spinach (S4) and earthy aroma (S6), a bitter (S9), earthy (S16) taste and a spinach (S22), earthy (S25) aftertaste together with an astringent mouthfeel (S26). It is the only tea that scored high on bitterness and astringency, attributes, which could contribute to consumers being averse to it. However, blending it with bush tea could act as a modulating factor. From the present study, the major aroma-active volatile compound found in the quadrant between bush tea and the blend of bush tea and special tea and which possess a medicinal, herbal, bitter and astringent taste included (-)-carvone (spearmint-like herbal). Special tea is associated with phytol (sweet, floral), α-copaene (woody), piperonal (cherry, vanilla, sweet anisic)**,** dihydroactinidolide (floral, rose-like) which does not exhibit the descriptive sensory attribute of bitter taste described in special tea. However, special attention should be given to the characterisation of volatile compounds in this tea since no data are currently available. Furthermore, tea fermentation, as well as blending special tea at different ratios with honeybush and rooibos tea, which exhibited a sweet taste, will help with reducing the bitterness. 

The volatile compounds closely linked to bush tea and special tea were 2-methyl butyraldehyde (fruity) and (-)-carvone (spearmint-like, herbal), respectively. The sensory attributes associated with bush tea were green and herbal aroma, green and herbal taste, green and herbal aftertaste. While (-)-carvone shares the herbal sensory attributes, 2-methyl butyraldehyde is known to exhibit a fruity attribute [39]. The poor correlation might have been attributed to the fact that many volatile compounds are responsible for a particular flavour sensation [44]. Furthermore, GC-MS data output revealed multiple compounds which might not have been further discussed due to the lack of previous literature or low cut-off similarity values. Bush tea and all its blends contributed 25% or less to the variation, indicating a modifying factor when different teas are combined. 

## 4. Conclusions

From the results, it is clear that the two indigenous bush teas (bush tea and special tea) and the blend of these two teas were characterised by a bitter flavour characteristic. Bush teas also had a strong green-cut grass, minty, dry green herbal-chai tea and medicinal aroma and flavour characteristic. Rooibos and honeybush teas were characterised by a stronger rooibos, sweet caramel, maple syrup and sweet typical honey aroma and flavours. Honeybush tea and the honeybush-bush tea blend were characterised by the strongest perfume aroma. The toning down of aversive flavour components in bush tea by the combination of honey and rooibos was a positive attribute and could serve as a motivation for the commercialisation of bush tea. 

Although volatile compounds, namely 2-furanmethanol, d-limonene, phenylethyl alcohol, linalool, geraniol, apocynin and α-myrcene, were found in honeybush tea, together with sweet aroma and woody cinnamon, it is still difficult to conclusively assume that the compounds mentioned are responsible for the descriptive sensory attributes. Some compounds which are responsible for a sweet, floral sensory flavour, such as α-copaene, piperonal, phytol, dihydroactinidolide, and 2-methyl butyraldehyde were also found in special tea and/or bush tea, which have bitterness and medicinal characteristics. More targeted compound identification could provide conclusive data on which compound is responsible for the bitter taste of bush and special tea. 

## Figures and Tables

**Figure 1 foods-09-00496-f001:**
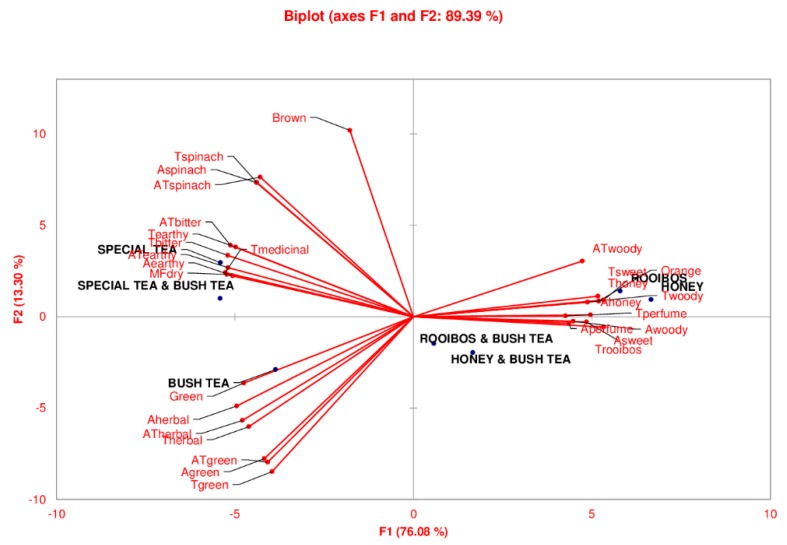
Principal component analysis (PCA) bi-plot showing distinct differences between the tea samples. The letters “A”, “T”, “A” and “MF” in front of an attribute name refer to aroma, taste, aftertaste and mouthfeel attributes, respectively.

**Figure 2 foods-09-00496-f002:**
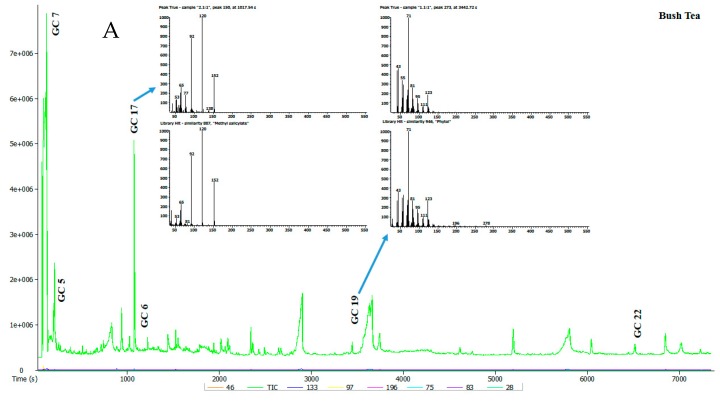
(**A**,**B**) Typical total ion chromatogram of the volatile compounds in selected tea samples.

**Figure 3 foods-09-00496-f003:**
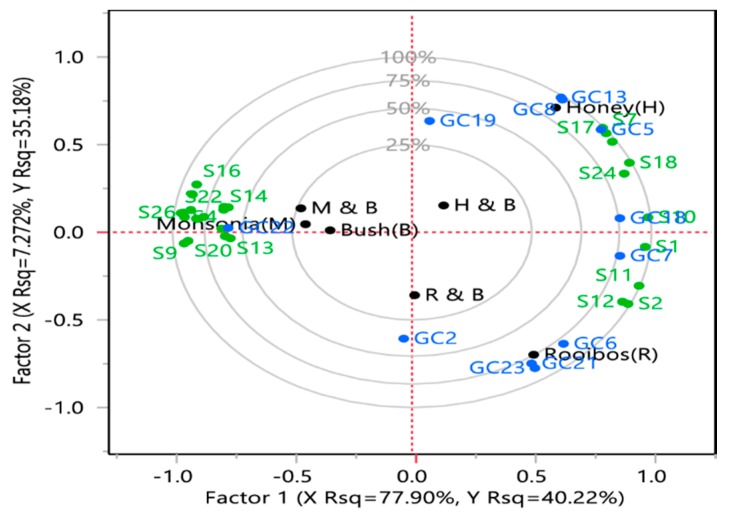
Partial least square regression analysis of descriptive sensory and gas chromatography data. Selected teas: bush tea (B), honeybush tea (H), (C) special tea (M), rooibos (R), 50% honeybush tea plus 50% bush tea (H and B), 50% special tea plus 50% bush tea and 50% rooibos tea plus 50% bush tea (R and B). Tentative identification of compounds: GC2, (E)-1-(2,3,6-trimethylphenyl)buta-1,3-diene (TPB, 1); GC5, 2-Furanmethanol; GC6, 2-Methoxy-4-vinylphenol; GC7, Acetic acid; GC8, α-myrcene; GC13, D-Limonene; GC16, Linalool; GC17, Methyl salicylate; GC18, Phenylethyl Alcohol; GC19, Phytol; GC30, Ocimene; GC32, 2-Methyl butyraldehyde; GC37, Dihydroactinidolide; GC41, Terpinolene; GC53, (E,E)-2,4-heptadienal; GC59, Maltol. Sensory data: S1, Asweet; S2, Ahoney, S3, Agreen; S4, Aspinach, S5, Aherbal; S6, A, Aearthy; S7, Afloral; S8, Awoody; S9, Tbitter; S10, TSweet; S11, Honey; S12, Trooibos; S13, Tgreen; S14, Tspinach; S15. Therbal; S16, Tearthy; S17, Tperfume; S18, Twoody; S19, Tmedicinal; S20, ATbitter; S21, ATgreen; S22, ATspinach; S23, ATherbal; S24, ATwoody; S25, ATearthy; S26, MFdry.

**Table 1 foods-09-00496-t001:** Descriptions of sensory attributes of teas.

Attributes	Description
**Aroma**	
Sweet—caramel, maple syrup	Aromatics associated with materials that also have a sweet taste, such as molasses, caramelised sugar and maple syrup.
Honey-sweet typical honey	Aromatics associated with the sweet, caramelised flora and woody aromatic associated with honey.
Green—cut grass, mint	Aromatics associated with green cut grass, fresh-cut grass, mint.
Cooked spinach	Aromatics associated with cooked spinach.
Dry green herbal—chai tea	Aromatics associated with “Green” flavour typical of dried grass or dried herbs.
Earthy—boiled potatoes, damp potting soil	Aromatics associated with damp soil, wet foliage, and damp potting soil.
Perfume—floral, lavender	Aromatics associated with having a light fragrant aromatic characteristic of lavender.
Woody cinnamon, dry dusty, bark	Aromatics associated with dry fresh-cut wood; bark, cinnamon, dust.
**Taste-Flavour**	
Bitter—quinine, caffeine	Flavours associated with the taste on the tongue stimulated by solutions of caffeine, quinine.
Sweet—caramel, maple syrup	Flavours associated with materials that also have a sweet taste, such as molasses, caramelised sugar, and maple syrup.
Honey—sweet typical honey	Flavours associated with the sweet, caramelised flora and woody aromatic associated with honey.
Rooibos	Flavours associated with a combination of honey, woody and herbal-floral notes with a slightly sweet taste and subtle astringency.
Green—cut grass, mint	Flavours associated with green cut grass, fresh-cut grass, mint.
Cooked spinach	Flavours associated with cooked spinach.
Dry green herbal—chai tea	Flavours associated with “Green” flavour typical of dried grass or dried herbs.
Earthy—boiled potatoes, damp potting soil	Flavours associated with damp soil, wet foliage, or slightly undercooked boiled potato, damp potting soil.
Fruity—peach, mango-like	Flavours associated with a mixture of peach-mango like fruits.
Perfume—floral, lavender	Flavours associated with a light fragrant aromatic characteristic of lavender.
Woody cinnamon, dry dusty, bark	Flavours associated with dry fresh-cut wood; bark, cinnamon, dust.
Medicinal	Flavours associated with dried grass or dried herbs used in herbal medication.
**Aftertaste**	
Bitter	Aftertaste associated with the taste on the tongue stimulated by solutions of caffeine, quinine.
Green—cut grass, mint	Aftertaste associated with green cut grass, fresh-cut grass, mint.
Cooked spinach	Aftertaste associated with cooked spinach.
Dry green herbal—chai tea	Aftertaste associated with “Green” flavour typical of dried grass or dried herbs.
Woody—cinnamon, dry dusty, bark	Aftertaste associated with dry fresh-cut wood, bark, cinnamon, dust.
Earthy—boiled potatoes, damp potting soil	Aftertaste associated with damp soil, wet foliage, or slightly undercooked boiled potato, or damp potting soil.
**Mouthfeel**	
Astringent/dry	The chemical feeling factor on the tongue or other skin surfaces of the oral cavity described as puckering/dry and associated with tannins or alum (unripe banana, strong tea, anise, allspice)

**Table 2 foods-09-00496-t002:** Mean values and significant differences in aroma, taste, aftertaste, mouthfeel and colour of the seven selected teas.

AROMA	CODE 1	CODE 2	*p*-Value	Intensity of Aroma, Taste, Aftertaste, Mouthfeel and Colour in Tea Samples						
**AROMA**				**Bush**	**Honeybush**	**Special**	**Rooibos**	**Bush and honeybush**	**Bush Tea and Special Tea**	**Bush and Rooibos**
Sweet—caramel, maple syrup	Asweety	S1	<0.0001	1.35 ^bc^	3.15 ^a^	1.25 ^cd^	3.13 ^a^	2.67 ^a^	1.30 ^cd^	2.52 ^a^
Honey—sweet typical honey	Ahoney	S2	<0.0001	1.18 ^cd^	2.81 ^b^	1.00 ^d^	3.98 ^a^	2.40 ^bf^	1.12 ^dg^	2.80 ^be^
Green-cut grass	Agreen	S3	<0.0001	4.22 ^be^	1.60 ^a^	2.67 ^ce^	1.43 ^a^	3.20 ^cd^	3.9 ^bd^	3.55 ^de^
Cooked spinach	Aspinach	S4	<0.0001	1.85 ^c^	1.00 ^b^	5.20 ^d^	1.00 ^a^	1.12 ^ab^	4.75 ^ed^	1.05 ^ab^
Dry green herbal—chai tea	Aherbal	S5	<0.0001	3.95 ^bc^	1.65 ^a^	3.38 ^cd^	1.30 ^a^	2.95 ^a^	3.45 ^bd^	2.95 ^a^
Earthy—boiled potatoes, damp potting soil	Aearthy	S6	<0.0001	2.02 ^bc^	1.25 ^a^	2.17 ^c^	1.07 ^a^	1.25 ^a^	2.12 ^cd^	1.30 ^a^
Perfume—floral, lavender	Aperfume	S7	<0.0001	1.18 ^ac^	4.62 ^b^	1.05 ^ce^	2.00 ^a^	3.02 ^d^	1.20 ^ae^	1.77 ^ac^
Woody cinnamon, dry dusty, bark	Awoody	S8	0.0075	1.48 ^b^	2.25 ^a^	1.32 ^b^	1.73 ^ab^	1.85 ^ab^	1.50 ^b^	1.48 ^b^
**TASTE**										
Bitter—quinine, caffeine	Tbitter	S9	<0.0001	4.47 ^bd^	2.73 ^a^	5.70 ^ce^	3.25 ^a^	3.58 ^ad^	5.33 ^e^	4.00 ^d^
Sweet—caramel, maple syrup	Tsweet	S10	<0.0001	1.18 ^b^	3.05 ^a^	1.15 ^cf^	2.77 ^ae^	2.38 ^de^	1.23 ^f^	1.80 ^bd^
Honey—sweet typical honey	Thoney	S11	<0.0001	1.10 ^bc^	2.58 ^a^	1.05 ^c^	3.25 ^a^	2.00 ^ad^	1.05 ^ae^	2.12 ^cd^
Rooibos	Trooibos	S12	<0.0001	1.20 ^cd^	3.45 ^be^	1.23 ^dg^	4.95 ^a^	3.45 ^f^	1.45 ^cg^	3.75 ^ef^
Green-cut grass, mint	Tgreen	S13	<0.0001	4.58 ^cd^	1.57 ^be^	2.83 ^dg^	1.52 ^a^	3.55 ^bf^	3.95 ^cg^	3.98 ^ef^
Cooked spinach	Tspinach	S14	<0.0001	1.77 ^bd^	1.00 ^a^	5.30 ^ce^	1.00 ^a^	1.10 ^be^	4.92 ^bf^	1.30 ^bd^
Dry green herbal—chai tea	Therbal	S15	<0.0001	4.20 ^b^	1.75 ^a^	3.52 ^cd^	1.32 ^a^	3.52^ab^	3.90 ^d^	3.73 ^ab^
Earthy—boiled potatoes, damp potting soil	Tearthy	S16	<0.0001	1.77 ^bc^	1.32 ^a^	2.23 ^cd^	1.12 ^a^	1.45 ^ce^	2.05 ^cf^	1.35 ^cd^
Perfume—floral, lavender	Tperfume	S17	<0.0001	1.18 ^bcf^	4.55 ^ab^	1.02 ^ce^	1.92 ^a^	2.80 ^af^	1.30 ^de^	1.62 ^ba^
Woody cinnamon, dry dusty, bark	Twoody	S18	0.0002	1.43 ^cd^	2.40 ^b^	1.45 ^cd^	1.82 ^a^	1.85 ^e^	1.32 ^ad^	1.65 ^ac^
Medicinal	Tmedicinal	S19	<0.0001	2.52 ^cef^	1.25 ^a^	3.42 ^d^	1.10 ^a^	1.82 ^ac^	3.33 ^df^	2.15 ^ce^
**AFTERTASTE**										
Bitter	ATbitter	S20	<0.0001	4.22 ^bd^	2.60 ^a^	5.67 ^cf^	3.15 ^ae^	3.55 ^be^	5.28 ^f^	3.80 ^de^
Green-cut grass, mint	ATgreen	S21	<0.0001	3.85 ^ceg^	1.52 ^b^	2.60 ^df^	1.43 ^a^	3.08 ^cdf^	3.50 ^efg^	3.50 ^de^
Cooked spinach	ATspinach	S22	<0.0001	1.55 ^a^	1.00 ^a^	4.60 ^b^	1.00 ^a^	1.07 ^a^	4.22 ^bc^	1.05 ^a^
Dry green herbal—chai tea	ATherbal	S23	<0.0001	3.92 ^bc^	1.55 ^a^	3.23 ^cd^	1.30 ^a^	3.15 ^bce^	3.67 ^def^	3.23 ^bd^
Woody cinnamon, dry dusty, bark	ATwoody	S24	0.0016	1.35 ^bdf^	2.25 ^a^	1.38 ^bc^	1.80 ^abc^	1.52 ^cde^	1.43 ^cf^	1.57 ^cd^
Earthy—boiled potatoes, damp potting soil	ATearthy	S25	0.0001	1.60 ^ab^	1.18 ^a^	1.82 ^b^	1.15 ^a^	1.27 ^a^	1.92 ^bc^	1.45 ^abc^
**MOUTH FEEL**										
Astringent/dry	MFdry	S26	<0.0001	4.42 ^bd^	3.05^a^	5.40 ^cf^	3.08 ^a^	4.03 ^be^	5.05 ^bf^	3.77 ^de^
COLOUR										
Brown	Brown		<0.0001	1.90 ^b^	3.20^af^	6.05 ^ce^	4.20 ^fg^	2.45 ^ab^	5.00 ^def^	4.65 ^df^
Orange	Orange		<0.0001	1.55 ^bc^	4.70^ae^	1.75 ^cd^	5.35 ^e^	3.65 ^a^	1.80 ^bd^	4.10 ^a^
Green	Green		<0.0001	5.70 ^b^	1.15^a^	4.10 ^cd^	1.25 ^a^	3.25 ^de^	4.35 ^ce^	1.85 ^a^

^a,b,c,d,e,f,g^ different superscript letters signify statistically significant differences in the relative intensity of each tea.

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
