# Peer review of "Sensory Characteristics and Volatile Compounds of Herbal Teas and Mixtures of Bush Tea with Other Selected Herbal Teas of South Africa"

_foods, 2020, doi:10.3390/foods9040496_

Round 1

Reviewer 1 Report

The manuscript presents the sensory characteristics and volatile compounds of herbal teas and mixtures of teas of South Africa. It is an interesting topic characterized by novelty. However, although the content of the article is well organized and well presented, there are some major issues that should be corrected prior to publication, especially regarding the part of the GC-MS analysis.

Regarding the article title I suggest to change the term “a mixture” with “mixtures”, since more than one mixture was reported.

In the abstract please replace the word “analyzed” with “performed”.

There is no mention in the manuscript regarding the origin of the chemicals (methanol, dichloromethane etc.) that was used in this work. Please add what it is necessary.

Line 90: The authors mention that “Bush tea was blended with the other three teas at a ratio of 1:1 (v/v).”, however until that point they only mention “tea powders”. Therefore, I think that it is confusing whether they mixed the powders or the extracts. Did mixing of the powder took place? If yes, do you mean (w/w) instead of (v/v)?

Line 78 and 101: It would be better to use the term “local market” instead of the full name of the supermarket.

Line 105: Please replace “briefly” with “Briefly”.

Table 1: Please remove the “-“ after Rooibos taste-flavor attribute.

Part 2.3.1. No information about the injection of the sample to the GC system is mentioned. Was it a liquid or headspace injection? If it was a liquid, add this information and include the sample quantity. If it was a headspace, add this information and include the sample equilibration/injection parameters.

Part 2.3.2. A lot of information about the detector is missing. Please add the model of the MS (The 7890B gas chromatograph is mentioned, whereas the model of the mass spectrometer is not mentioned). Also please add the m/z range of the recorded ions, the MS transfer line temperature and the MS quadrupole temperature (if applicable) etc. Moreover, add information regarding the column supplier is missing.

Table 2: It would be better to replace the “SELECTED TEA SAMPLE” with a term that points out what the numbers in the cells are. Also, explain the use of a,b,c,d,e,f.

.

Line 252-254:  Please delete the sentence “From the chromatograph, it is evident that the chemistry of the compounds versus that of the column determined the order of the elution.”

Line 260: “Linalool” is mentioned while in line 262 “cis of linalool” is mentioned. Do the authors mean a derivative of linalool? Linalool has the (R)-(−)-linalool and (S)-(+)-linalool enantiomers, however stereoisomers (cis-trans isomers) may refer to a chemical derivative of linalool and not the compound as such. Please check this throughout the manuscript since “cis of linalool” is mentioned in many points.

Figure 2. The quality of the figure is very low and the mass spectra are not visible, please improve the quality.

Table 3:

First of all in some cells the authors use “,” instead of “.” for decimal points. Please check the entire table and replace it.

Moreover, for the RT please add the unit and use minutes instead of seconds for the retention time.

It would be better to report the compounds based on their retention time.

In line 252 it is mentioned that peak identification was also based on the linear retention index (LRI) values. Please include in the table the experimental values and the values from the literature for each compound.

All the compounds of the table should be checked again. Terms as “ç-Terpinene” “Cis of linalool”, “trans-á-Ocimene” and “ionene” are wrong.

Please replace the term “d-limonene terpene” with “d-limonene” throughout the manuscript

Is the peak area used in table 3 an average value? Was it based on triplicate analysis? There is no mention regarding that.

Moreover, there is no point to mention the peak area in the table. It could be more useful to normalize the peak areas for each sample and report the % value for each compound. Also it would be better to make an extra line to add the varieties of the tea there instead of the first line and in the first line “Normalized Peak Area” should be placed.

Please explain better the use of a,b,c in this table.

Reviewer 2 Report

Manuscript Foods_720504

“Sensory characteristics and volatile compounds of herbal teas and a mixture of bush tea with other selected herbal teas of South Africa”

# Reviewer 1

I cannot review the paperbecause the procedure used by the authors to quantify volatile molecules is not clear.  In the materials and methods section, no procedure is indicated and in Table n.3, I observe the areas not normalized. The quantification in GC-MS can be performed in different ways: external calibration, internal standard method, determination of percentage ratios, etc (de Oliveira et al. 2010doi.org/10.1590/S0100-40422010000400041). According to de Oliveira et al. (2010) I think that is very important to use for example the internal standard standardization because this is done to correct analyte losses during sample preparation.

If this very important aspect is not clarified, I cannot proceed with the revision.I await guidance from the authors and then re-evaluate the review.

Round 2

Reviewer 1 Report

The authors used most of the comments in order to improve the quality of the manuscript. We appreciate the effort of the authors. However, there are still some issues that have to be addressed prior to publication.

Comments 12 - Line 260: "Linalool" is mentioned while in line 262 "cis of linalool" is mentioned. Do the authors mean a derivative of linalool? Linalool has the (R)-(−)-linalool and (S)-(+)-linalool enantiomers, however stereoisomers (cis-trans isomers) may refer to a chemical derivative of linalool and not the compound as such. Please check this throughout the manuscript since "cis of linalool" is mentioned in many points

Author response: We appreciate the reviewer's comments; however, the library used to identify the compound does so based on the fragmentation pattern of the compound and does not give the stereo-chemistry of the compound. The isomers R and S cannot be differentiated.

Reviewer response: The authors mentioned correctly that linalool has two enantiomers: R and S. However, in the manuscript they use the term “cis of linalool”. Linalool does not have stereoisomers and therefore, “cis of linalool” is not a correct term. Please explain which chemical compound you mean. Is it a derivative of linalool, for example an oxide?

Comments 15: Moreover, for the RT please add the unit and use minutes instead of seconds for the retention time.

Author response: The retention time is generated by the software, changing them to minutes might introduce an error. 

Reviewer response: Please convert the RT to minutes without introducing an error. There is no point in using seconds as retention time in gas chromatography.

Comment 17: In line 252 it is mentioned that peak identification was also based on the linear retention index (LRI) values. Please include in the table the experimental values and the values from the literature for each compound.

Author response: The comparison was based on the fact that it was reported in other literature. Not that they came at a similar retention time. The retention time might differ depending on the column used. The sentence mentioned in the comment has been removed from the text.

 Reviewer response: Of course retention time differ based on the column and the oven program, however linear retention index (LRI) values is a way to compare different instrumentations. There is a huge difference between “retention time” and “retention index”.

Comment 18: All the compounds of the table should be checked again. Terms as "ç-Terpinene" "Cis of linalool", "trans-á-Ocimene" and "ionene" are wrong.

Author response: The four identified compound has been removed from the manuscripts. However, the four compounds that are removed will still be validated in our future trials.

Reviewer response: The point was to provide the correct names and not remove the compounds. There are still errors throughout the manuscript. The authors still use the term “Cis of linalool” which chemically makes no sense.

Comment 21: Moreover, there is no point to mention the peak area in the table. It could be more useful to normalise the peak areas for each sample and report the % value for each compound. Also, it would be better to make an extra line to add the varieties of the tea there instead of the first line and in the first line "Normalized Peak Area" should be placed.

Author response: The comment from the reviewer is highly appreciated; the point will be taken as a suggestion to be incorporated in future work.

Reviewer response: If you intend to keep the peak areas please indicate the reason, since peak areas are just “raw data” and not something meaningful.

Moreover, since you mentioned that every analysis was performed in triplicate, it is necessary to add the standard deviation to each value in the table.

Reviewer 2 Report

Reviewer #1

Line 152:  Liquid or gas chromatography??? Please correct it;

Line 172: Insert the following sentence “Samples for GC-MS were prepared and analysed in triplicate” in the “2.3.1. Tea samples” paragraph;

Line 284: The authors reported that they have identified 63 molecules but in Table 3 are reported 59 molecules. How many molecules have been identified?

Line 288-289:The authors affirmed that: The identified volatile compounds and their relative area, mean values and standard error are summarised in Table 3” ..but in Table 3 I only see the peakareas and the anova symbols.
